# Assessing and Predicting Green Credit Risk in the Paper Industry

**DOI:** 10.3390/ijerph192215373

**Published:** 2022-11-21

**Authors:** Yue Zhao, Yan Chen

**Affiliations:** 1College of Economics and Management, Nanjing Forestry University, Nanjing 210037, China; 2Academy of Chinese Ecological Progress and Forestry Development Studies, Nanjing Forestry University, Nanjing 210037, China

**Keywords:** green credit risk, paper industry, random forest, KMV model, Gini index

## Abstract

The paper industry is closely related to forestry resources, which constitute an essential part of achieving sustainable development. Green credit can provide financial support to assist the paper industry in achieving carbon neutrality. To develop a method for performing green credit risk assessments in the paper industry, first, an initial index system was established on the basis of two dimensions: financial risk and socio-environmental risk. Then, the KMV model was applied to measure credit risk. The combined results of this model, along with the environmental penalties of an enterprise, formed the basis for the classification of green credit risk. Third, the Gini index was used to filter out, one by one, the indexes with the least influence among the factors, and then random forest iterations were performed until the prediction accuracy reached the optimum, thus establishing a green credit risk prediction model for the paper industry. The results show that the accuracy of the sample classification reached 93.75%, and the accuracy of the sample classification for high-risk enterprises reached 100%. The established index system offers good guidance for the assessment of green credit risk in the paper industry, in which the interest coverage ratio, current ratio, asset-liability ratio, and green emissions are the main factors affecting green credit risk.

## 1. Introduction

Nowadays, much more attention is paid to environmental issues in the pursuit of economic growth [1,2]. To speed up reform, green credit will be an important aspect of promoting and achieving ecological progress. Highly polluting industries cause great damage to the environment. It is necessary for the traditional paper industry to cut down a large number of trees, destroying the balance of the original ecosystem, while the paper-making process is accompanied by pollution such as sewage and odors. Therefore, industrially upgrading the paper industry is an imminent necessity. Industrial upgrade and green credit are closely linked, as the paper industry requires the support of green credit in order to be able to carry out technological innovation. The green trade barrier refers to a series of import restriction measures formulated by importing countries in order to protect natural resources, the ecological environment, and human health when carrying out activities related to international trade. As a traditional industry with high pollution and high energy consumption, if the level of green paper-making technology employed by enterprises remains relatively low, these enterprises could easily suffer negative consequences due to green trade barriers, which will not be conducive to the future development of the industry. The paper industry also needs the support of green credit in order to be able to implement policy. China launched “the combination project of forest and paper” policy with the aim of developing commercial forests while taking forest carbon sequestration into account in order to achieve the strategic goal of sustainable development. Green credit risk assessments in the paper industry help commercial banks avoid the risk of default arising due to the imposition of environmental penalties on enterprises whose emissions do not meet environmental requirements. It is helpful to provide financial support for genuine environmental projects and green transformation in the paper industry, as these not only enhance the profitability of commercial banks in terms of green credit, but also promote the flourishing of green industries, assisting with the realization of the goal of sustainable development.

The management of credit risk in banks involves two main aspects: on the one hand, the systemic risk faced by the bank in the course of its own operation, and on the other hand, the credit risk generated by the bank in the course of its operation as a credit business. There are many different methods for measuring systemic risk. Marginal expected shortfall (MES) is often used as an index for measuring systemic risk, and this index has been used to establish a model for risk prediction [3,4]. The SRISK function was constructed to make systemic risk assessment results more stable, such that they would not be affected by short-term market noise [5]. To measure the spillover effect of systemic risk, the widely used CoVaR model was constructed [6]. Domestic and foreign banks focus greatly on the assessment and prediction of credit risk. As a result, many scholars have begun to study credit risk. The Kealhofer, Mcquown and Vasicek (KMV) model, based on the Black–Scholes–Merton (BSM) model, is often used to monitor credit risk [7], and effectively predicts credit risk and reflects the default risk degree of an enterprise [8,9]. To improve risk resistance ability, scholars have used the generalized autoregressive conditional heteroskedasticity (GARCH) model and the threshold regression model to modify the important parameters of the KMV model, which was then used to assess the impact of changes caused by the COVID-19 pandemic on the credit risk of China’s healthcare industry [10]. The KMV model has also been used to study the impact of China’s insurance industry on credit risk, providing a reference for the formulation of financial stability policies [11]. The KMV model has been applied not only in the assessment of the credit risk of listed companies, but also in the assessment of government debt risk. The governments of all countries are facing an increasing number of financial risks and uncertainties [12]. Scholars have used the KMV model to measure government debt risks, and have studied and analyzed the government debt of emerging countries [13], Chinese government debt [14], and local government debt [15]. With the advent of the big data era, machine learning methods such as support vector machines (SVMs), artificial neural networks (ANNs), and the classification and regression tree (CART), which present few limitations and high data processing power, have gradually begun to be applied in the field of credit risk assessment [16,17,18]. Some scholars have combined the correlation vector machine and the decision tree to form a decision support model (EDSM), and used multicriteria decision-making methods to determine early warning mechanisms, thus improving the accuracy of credit risk assessments, overcoming issues related to timeliness, and reducing the cost of credit risk [19]. To overcome the failures of the fuzzy comprehensive evaluation in the credit risk evaluation of commercial banks, some scholars have built risk evaluation models based on the Hopfield neural network, which can reflect the current credit risk status [20]. Scholars have extended credit risk assessment research to different fields. To alleviate the financing difficulties of small and medium-sized enterprises (SMEs), the credit risk prediction of supply chain finance has become the key to making financing decisions [21,22,23]. With the increasing proportion of personal loans, personal credit risk has been rising, whether for bank loans or peer-to-peer (P2P) network loans. Thus, scholars have conducted an individual credit risk assessment to reduce personal credit risk [24,25]. Some scholars improved the BP neural network model and constructed an integrated IDGSO-BP model for micro- and small enterprises that can effectively improve the accuracy of credit risk assessments [26].

Green finance integrates environmental protection and economic interests [27]. In China, banks use corporate environmental protection as one of the bases for extending green credit. It not only improves the net profit of commercial banks [28], but also reduces the credit risk [29] and environmental pollution risk [30] of commercial banks. Because of the gradual expansion of green credit in China, many scholars have researched green credit risk assessment. Some scholars found that the risks of the wind power and photovoltaics industries were similar to those of traditional industries, but the risk gap between companies was more extensive [31]. To address the issue of innovation benefits and risks arising from green credit in commercial banks, the grey relational analysis (GRA) was used to find that more than half of the banks studied developed green credit with higher benefits and lower risks than the traditional credit business [32]. An improved KMV model was used to study the green credit risk in the new-energy vehicle industry and its internal subsectors. It was found that the risk of the new-energy vehicle industry was similar to that of traditional industries. Still, there were differences between subsectors [33]. Some scholars quantitatively analyzed the green credit risk of listed companies in different sectors based on the backpropagation neural network (BPNN) model. It was found that medium- to high-risk companies accounted for 11.5%, and most companies in the coal industry faced a higher green credit risk [34].

According to previous studies, scholars have conducted deep research on credit risk assessments, using artificial intelligence, big data, and other methods to perfect them. Later, the credit risk assessment method was applied to the field of green credit, where green factors have not been really considered. The green credit research fields are mostly in low-polluting and low-energy-consuming sectors, such as the new-energy industry, and there is no research on high-polluting and high-energy-consuming sectors, such as the paper industry. Therefore, the characteristics and contributions of this paper are: Firstly, the index system not only takes the green factors into account, but also has paper industry specificity. Considering the paper industry’s financial and socio-environmental responsibility, a set of green credit risk evaluation index systems are established. Secondly, compared to other models, the random forest model has a higher accuracy in data mining. This paper combines the random forest with the KMV model, which is widely used in credit risk assessments, to establish the assessment model.

## 2. Materials and Methods

This study was divided into four steps (see Figure 1). The first step was to set the initial evaluation index system for the green credit risk of listed paper-making enterprises as the input value of the random forest. The initial evaluation index system included two dimensions: the financial index and the socio-environment index. The second step was to classify the green credit risk for enterprises based on the KMV model. The combined results of this model, along with the environmental penalties of an enterprise, formed the basis for the classification of green credit risk. The third step was to conduct random forest iterations. The Gini index was used to continuously screen out the least significant indicators to explore the main factors affecting the green credit risk of the paper industry. The fourth step was to establish the paper industry’s final green credit risk assessment prediction model.

### 2.1. Construct the Initial Evaluation Index System

A reasonable evaluation index system is the basis of the green credit risk assessment, whereas a practical and professional evaluation index system can improve the accuracy and validity of the evaluation results. The green credit implementation regulations of major Chinese commercial banks and the Green Credit Guidelines issued by the China Banking Regulatory Commission (CBRC) in 2012 indicate that the green credit risk assessment index system should not only involve the financial status of enterprises, but also take social and environmental risks into account. Moreover, profitability is related to environmental performance [35], and the assessment of green credit risk can consider more than just the financial performance of enterprises. Therefore, based on the physical truth, this paper established a green credit risk assessment index system for the paper industry to provide a comprehensive response to corporate green credit risk from the financial and socio-environmental perspectives.

This paper selected solvency, profitability, operating capacity, and growth ability as the primary indexes to measure the financial credit risk of listed enterprises [36,37]. Each dimension selected four related subindicators as secondary indexes. Solvency is the main index to determine whether an enterprise can repay debts. We divided the solvency into four secondary indexes: the current ratio, quick ratio, asset-liability ratio, and interest coverage ratio. The current ratio, quick ratio, and asset-liability ratio were used to measure the liquidity of the company’s assets, and the interest coverage ratio was used to measure the company’s ability to pay interest. Profitability is the main index to determine the efficiency of debt repayment. We divided the profitability into four secondary indexes: ROE, ROA, net asset value per share, and net interest rate in sales. Operating capacity is the main index that determines the efficiency of the capital operation of an enterprise. We divided the operating capacity into four secondary indexes: inventory turnover, total assets turnover, accounts receivable turnover, and current assets turnover. Growth capability is the main index that determines the future development capability of an enterprise. We divided the growth capability into four secondary indexes: the revenue growth ratio, net profit growth ratio, net asset growth ratio, and EPS growth rate.

We measured the socio-environmental index from two perspectives: social responsibility and environmental performance. Enterprise social responsibility was measured by the social responsibility score published by Hexun.com. The Hexun Social Responsibility Score for listed companies is the score that Hexun calculates by systematically evaluating corporate social responsibility undertakings through a system of relevant indicators, with scores published on Hexun.com. There is a strong correlation between enterprise environmental disclosure and environmental pollution [38]. Therefore, we divided the environmental performance index into two secondary indexes: green culture and green emissions. The connotation of green culture is rich, and thus, it is impossible to define green culture accurately. In this paper, the keywords appearing in corporate social responsibility reports (CSRs) or annual reports were used to represent enterprise green culture [39], which include social responsibility, energy saving, emission reduction, green ecology, sustainable development, biodiversity, low-carbon production, and recycling. Each keyword appearing in the report was assigned a score of 1. The maximum cumulative score was 8 points, and the minimum was 0 points.

The granting of green credit also needs to be based on the evaluation of the environmental protection situation of polluting industries, such as resource consumption, pollution emissions, and environmental disclosure, in their daily production and business activities. The better the environmental credentials of the enterprise, the higher the degree of pollution treatment. Therefore, this paper used the major pollution emissions of the industry as a proxy for environmental performance [40]. The main pollutant emissions from the paper industry include chemical oxygen demand (VC) and ammonia nitrogen (VN) in wastewater, and sulfur dioxide (VS) and nitrogen oxides (VO) in the exhaust gas. Therefore, VC, VN, VS and VO were included as measures when considering the green value of an enterprise. To exclude the influence of the size of listed companies on pollutant emissions, this paper used annual emissions of a particular pollutant divided by the company’s total operating revenue in the current year to represent unit emissions. After that, the entropy-weighted-TOPSIS [41] model was used to measure the green emission score of the sample.

According to the above process, the final green credit risk assessment index system was obtained under two dimensions of the financial and social environment (see Table 1), totaling 19 initial indexes. The index system not only avoided the subjectivity of qualitative indicators, but also made the selection of indicators more industry-specific than the traditional index system.

### 2.2. Enterprise Green Credit Risk Classification

This paper comprehensively considered credit risk and environmental risk to measure whether paper-making enterprises had a high potential for green credit risk.

#### 2.2.1. The Basis for Classifying Enterprise Green Credit Risk

In terms of credit risk, the KMV model was used to calculate the credit risk of enterprises in different years. We must select a standard as the main basis for classifying enterprise credit risk. That is, the paper-making enterprise did not have a high credit risk and was deemed qualified if the default distance (DD) was not lower than a certain number, whereas the paper-making enterprise had a high credit risk and was deemed unqualified if the DD was lower than a certain number. The focus of green credit origination also includes environmental risks. This paper decided whether there was an environmental pollution incident as the basis for “green” judgment. Reviewing the environmental penalty information of each paper-making enterprise from 2017 to 2021, if there were major environmental pollution incidents or fines, that means that the company lacks environmental protection and should be set as unqualified. If there were no major environmental pollution incidents or fines, that means that the company works well with regard to environmental protection and should be set as qualified. We used the combined results of the DD and environmental penalties to classify green credit risk for paper-making enterprises. The output value is 0 or 1. That is, the random forest output value is 1 if one of the paper-making company’s credit risk assessment and environmental risk assessment fails, and the random forest output value is 0 if the paper-making company passes both.

#### 2.2.2. Credit Risk Assessment Model

The KMV model has wide applicability and is often used in credit risk assessment. The model evaluates the credit risk of listed companies by calculating the expected default frequency (EDF) from the asset volatility and asset value. The discriminatory accuracy of EDF in China is inferior to that of DD [42]; therefore, DD is often used to judge the credit risk of listed companies. The credit risk assessment based on the KMV model is divided into three steps:

The first step is to figure out the asset value and the annual volatility of asset values. The key to the KMV model is to figure out each enterprise’s asset value and annual volatility of asset value through the enterprise equity value and annual volatility of equity value. The simultaneous equations are as Equation (1) shows:(1)$$\{\begin{array}{c} V_{E}=V_{A}N\left(d_{1}\right)-De^{-rT}N\left(d_{2}\right)\\ \sigma_{E}=\frac{V_{A}N\left(d_{1}\right)\sigma_{A}}{V_{E}}\; \end{array}$$

$V_{E}$ is the equity value of a listed enterprise, $\sigma_{E}$ is the equity value volatility of a listed enterprise, $V_{A}$ is the asset value of a listed enterprise, $\;\sigma_{A}$ is the asset value volatility of a listed enterprise, N(x) is the standard normal distribution, $d_{1}=\frac{\left[Ln\left(\frac{V_{A}}{D}\right)+\left(r+\frac{\sigma_{A}^{2}}{2}\right)\right]}{\sigma_{A}\sqrt{T}}$, $d_{2}=d_{1}-\sigma_{A}\sqrt{T}$*, D* is the enterprise book value, r is the risk-free rate, and *T* is the repayment term, all of which can be acquired from publicly available market data. The equity value volatility in Equation (1) is calculated using the historical volatility method, and the formula is as follows:(2)$$\sigma_{E}=\sigma_{n}*\sqrt{n}$$

The daily standard deviation of the stock is $\sigma_{n}=\sqrt{\frac{1}{n-1}\sum_{i=1}^{n}{\left(u_{i}-\bar{u}\right)}^{2}}$, the logarithmic return of the stock is $u_{i}$ = $Ln\left(\frac{S_{i}}{S_{i-1}}\right)$, and n is the number of days the stock of the listed company is traded in a year.

The second step is to figure out the default point (DP). By studying historical default data, KMV found that the default condition is only partially valid when the enterprise asset value is lower than the debt value due to long-term debt (LTD). Enterprises choose to default when the value of assets is less than half of the value of short-term debt (STD) plus the value of long-term debt. The formula is as follows:(3)DP=STD +50%LTD 

The third step is to figure out the value of the DD. The value of the DD represents the distance of the enterprise market asset value from the DP. Enterprises are less likely to default if the value of the DD is long, and enterprises are more likely to default if the value of the DD is short. Apply $\sigma_{A}$ and $V_{A}$, obtained by the Newton iteration method, to Equation (4):(4)$$\mathrm{DD}=\frac{V_{A}-\mathrm{DP}}{V_{A}\sigma_{A}}$$

### 2.3. Green Credit Risk Prediction Model

In this paper, we used the random forest model for green credit risk prediction in the paper industry. The initial evaluation index value was used as the input value of the random forest, and the conclusion of green credit risk classification was used as the output value of the random forest.

Random forest is a classification model that combines random sampling and decision trees. The indexes of all samples are entered into the random forest. After that, the sample size and indexes are randomly selected by the bootstrap sampling method to construct the decision tree models. Each decision tree model obtains a different classification result, and thus, the decision tree classifier produces a judgment with the principle of the minority obeying the majority to improve the prediction accuracy. The principle of the random forest algorithm is shown in Figure 2.

As a machine learning algorithm, the prediction accuracy of random forest can be further improved by iterative computation on the underlying model. The Gini index applied to the random forest model can discern the importance of the input indexes [43]. The index importance analysis can identify important variations to reduce the dimensions of the original index system, save computing costs, and improve prediction accuracy. The Gini index score determines the importance of each index in predicting green credit risk, and the larger the value that is indicated, the greater the importance. The specific calculation process is as follows:(5)$$GI_{q}^{\left(i\right)}=1-\sum_{C=1}^{\left|C\right|}{\left(P_{qc}^{\left(i\right)}\right)}^{2}$$

GI represents the Gini index; C represents the category; and $P_{qc}$ represents the proportion of category c in node q.
(6)$$VIM_{jq}^{\left(Gini\right)\left(i\right)}=GI_{q}^{\left(i\right)}+GI_{l}^{\left(i\right)}+GI_{r}^{\left(i\right)}$$

Equation (6) shows the importance of the index $\;X_{j}\;$ at node  q  of the ith tree. VIM represents the score of variable importance; I represents the number of decision trees; J  represents the indexes X1, X2, X3, …, XJ; and $GI_{l}^{\left(i\right)}$ and $GI_{r}^{\left(i\right)}$ represent the Gini index of two new nodes after branching.

Suppose Q is the ensemble of nodes where index $X_{j}$ appears in decision tree i; then, the importance of $X_{j}$ in the ith tree is shown in Equation (7):(7)$$VIM_{j}^{\left(Gini\right)\left(i\right)}=\sum_{q\in Q}VIM_{jq}^{\left(Gini\right)\left(i\right)}$$

Suppose there are  I trees in the random forest. Equation (8) can determine the VIM:(8)$$VIM_{j}^{\left(Gini\right)}=\sum_{i=1}^{I}VIM_{j}^{\left(Gini\right)\left(i\right)}$$

The normalized processing of VIM is calculated by Equation (9):(9)$$VIM_{j}^{\left(Gini\right)}=\frac{VIM_{j}^{\left(Gini\right)}}{\sum_{j^{\prime }=1}^{J}VIM_{j^{\prime }}^{\left(Gini\right)}}$$

### 2.4. Date Sources

We selected 26 A-share listed paper-making enterprises based on the research period between 2017 and 2021 for the study. The total number of A-share listed paper-making enterprises in China is 40. There was no mandatory requirement for enterprises to publish pollution emission data until 2022, and therefore, the published data were not comprehensive. This paper screened out the paper-making enterprises that were missing serious environmental information disclosures, and the remaining enterprises numbered 26. This paper selected 2017–2021 as the measurement interval because of the incomplete disclosure of corporate environment-related data before 2017. December 31 of each of the five years was used as the base date for the measurements. If there were paper-making companies listed after 2017, the year after listing was taken as the empirical interval.

The paper-making industry is a highly polluting industry, and its main pollutant emissions contain chemical oxygen demand (VC) and ammonia nitrogen (VN) in wastewater, and sulfur dioxide (VS) and nitrogen oxides (VO) in exhaust gases. As a result, we took the pollutants mentioned above as the influencing factors of green emissions. The annual reports and CSRs of each company from 2017 to 2021 were reviewed to obtain the emissions of chemical oxygen demand (VC), ammonia nitrogen (VN), sulfur dioxide (VS), and nitrogen oxides (VO). Then, we screened out the years with incomplete environmental information disclosures of 26 paper companies. Finally, we obtained 104 samples for the study. All financial data of listed enterprises came from the CSMAR, and pollution emission data came from corporate annual reports or CSRs.

## 3. Case Study

### 3.1. Case Study of Green Emissions

The green emission score of each enterprise was calculated with the help of MATLAB (see Table 2). On the one hand, the degree of environmental pollution levels of different paper-making enterprises varies greatly. The highest score was for Yuto Packaging Technology from 2019 to 2020, almost reaching a full score and indicating that the enterprise had light environmental pollution. The lowest score was for Yueyang Forest & Paper in 2017, with a score of 0.3717, indicating that the enterprise had serious environmental pollution. On the other hand, the green emission score fluctuated widely in different years, even for the same paper-making enterprise. The most volatile fluctuation was found for Songyang Recycle Resources, with a variance of 0.02792, indicating that the severity of environmental pollution was variable. The least volatile fluctuation was found for Yuto Packaging Technology, whose score was the top score over the past five years, indicating that the enterprise pays more attention to pollutant emissions.

### 3.2. Case Study of Enterprise Green Credit Risk Classification

#### 3.2.1. Parameter Settings for Enterprise Credit Risk Assessment

We evaluated credit risk using the KMV model with the following parameter settings:(1)Parameter setting for the risk-free rate (r).

In this paper, we decided to use the one-year fixed deposit interest rate published by the People’s Bank of China (PBC) to make up for the gap in China’s official risk-free rate data. If an interest rate adjustment was encountered during the year, the calculation formula would be $r=\sum_{1}^{n}r^{*}*\frac{the\;number\;of\;sequential\;days\;for\;r^{*}}{the\;number\;of\;days\;in\;the\;year}$. $r^{*}$ is the real interest rate announced by the People’s Bank of China.

(2)Parameter setting for the repayment term (T).

The repayment period was set as one fiscal year, T = 1.

(3)Parameter setting for the equity value ($V_{E}$).

In the complex situation that the shareholding reform has been completed and the stock market is mixed with “tradable shares” and “non-tradable shares” at the same time, this paper defined the equity value as follows: enterprise equity value $\left(V_{E}\right)$ = closing price of shares at the end of the year (p) * the number of tradable shares $\left(Q_{C}\right)$ + net assets per share (NAPS) * the number of non-tradable shares $\left(Q_{n}\right)$.

#### 3.2.2. Case Study of Credit Risk Assessment

The credit risk assessment of each enterprise was calculated with the help of MATLAB (see Table 3). Observing the default distance of each company, we can find that the greatest default distance was obtained by Forest Packing in 2020, with a distance of 5.1820, indicating a relatively low credit risk in that year. The least default distance was obtained by the Chenming Group in 2019, with a distance of −7.9213, indicating a relatively high credit risk in that year. The maximum difference in default distance between listed enterprises was 13.1033, indicating a large credit risk gap between enterprises. In terms of the default distance of the same enterprise over five years, Xianhe Corp. had the most volatile fluctuation in the default distance with a variance of 5.6656, indicating that the credit risk was unstable. Songyang Recycle Resources had the least volatile fluctuation in the default distance with a variance of 0.0287, indicating that the credit risk was stable.

First, we can find that most paper-making enterprises had a default distance between −2 and 3. Thus, the probable criteria for judging enterprise default should be between −2 and 3. Moreover, determining the average value of the default distance of these 104 samples was used to clarify the criteria for judging enterprise default. The mean value was 1. That is, if the default distance was less than 1, the credit risk of the paper-making enterprise should be unqualified. Meanwhile, the credit risk of the paper-making enterprise was determined to be qualified if the default distance was greater than or equal to 1.

### 3.3. Study Case of Green Credit Risk Prediction

#### 3.3.1. Parameter Settings for Random Forest

Firstly, we divided all samples into a training set and test set. Bootstrap sampling in the random forest model only contains 37% of the original samples, which means that about 67% will not appear in the training set. Therefore, the ratio of the training set to the test set is generally 7:3. In this paper, we put 72 samples into the training set (n_train), and 32 samples into the test set (n_test).

Secondly, we determined the sampling times (n_tree). The accuracy of the random forest improves to be stable with the increase in sampling times. Therefore, the sampling times were set to 3000 in this paper (n_tree = 3000).

Finally, the number of indexes (m_features) needed to be adjusted, which is one of the most important parameters of the random forest. As the number of indexes increases, the accuracy of the random forest will also improve. However, if the number of indexes continues to rise, the correlation between decision trees will also increase. In other words, the accuracy of the random forest model will decrease as the number of indexes increases. Generally, the random forest complexity is moderate when $m\_\mathrm{features}=\frac{\sqrt{M}}{M}$. We assumed that the initial value of m_features was 0.3.

We used Python to construct the initial random forest algorithm based on the parameters set above. To figure out the optimal parameters given by printing the gradient optimization, we input 0.2, 0.3, and 0.4 into the random forest, which was carried out to find the initial value of m_features and its proximal minima. It can be seen that when m_features=0.4, the random forest model error rate of the test set was the smallest (12.5%), and the model accuracy was the highest (see Table 4).

The importance of the 19 indexes in the random forest was measured by the Gini index (see Figure 3).

X18 had the least influence on the model and was removed according to the importance ranking, and then, the remaining 18 indexes were used to train the random forest again until the misclassification rate was minimized. After the above steps, we can find that the error rate of the model was the lowest when the number of indexes decreased to 11, and at this point, m_features = 0.2 (see Table 5).

In the end, the minimum misclassification rate of the random forest was 6.25%, which meant that the prediction accuracy reached 93.75%. The model covered 11 indexes (see Figure 4), which are, in order of importance, the interest coverage ratio (X04), current ratio (X02), asset-liability ratio (X01), green emissions (X19), inventory turnover (X09), revenue growth ratio (X13), net profit growth ratio (X14), ROE (X05), EPS growth ratio (X16), current assets turnover (X12), and total assets turnover (X10).

We set the socio-environmental indexes from the perspective of the degree of corporate social responsibility (social responsibility score), environmental awareness (green culture), and actual environmental protection actions (green emissions). After the random forest iteration, green emissions was the only one left, which meant that enterprises’ actual environmental protection actions played a more important role in the classification of green credit risk.

#### 3.3.2. Robustness Test

This paper had assumed that a DD less than 1 or enterprises subjected to environmental penalties had potential green credit risks. This assumption was adjusted to test the model’s robustness further. That is, a DD less than 0.5 or the enterprises subjected to environmental penalties had potential credit risks. Firstly, the variable parameter involved in the above assumption was set as i. Secondly, the green credit risk classification result obtained as i = 0.5 was substituted into the random forest model. Finally, the above model construction and iteration process was repeated. The results show that when the misclassification rate of the random forest drops to a low level, the final indexes of the model include the interest coverage ratio, current ratio, asset-liability ratio, green emissions, inventory turnover, current assets turnover, revenue growth ratio, quick ratio, net profit growth ratio, total assets turnover, EPS growth ratio, and other indicators, but the order of importance was slightly adjusted. It can be concluded that the above indicators were exactly the main factors which affected the green credit risk (see Table 6), and the previously established random forest model was robust.

### 3.4. Discussion of Results

(1)The DD calculated by the KMV model showed the credit difference between paper enterprises, which can distinguish the credit risk status faced by other paper enterprises. Even the DD of the same paper enterprise in different years could fluctuate sharply; thus, enterprises should strengthen the stability of their risk management.(2)The random forest model had a high prediction accuracy for the green credit risk assessment of the paper industry, as shown in Table 5. The prediction accuracy of the model on the test set reached 93.75%. This indicates that the random forest can effectively assess the green credit risk of paper-making enterprises, which further proves the scientificity of the index system and the effectiveness of the assessment model constructed in this paper.(3)Commercial banks will decide whether to extend green credit based on the forecast results. Generally speaking, commercial banks will suffer a greater loss when they extend credit to enterprises with a high credit risk and normal prediction results, compared with enterprises with a normal credit risk and high prediction results. It can be seen from Table 5 that the correct rate for the classification of high green credit risk enterprises in the paper industry reached 100%. It was further shown that the green credit risk assessment model based on the random forest algorithm for paper enterprises was applicable, and can effectively prevent the loss of commercial banks because of misjudgment.(4)Figure 4 shows 11 indexes, including 5 primary indexes from two perspectives. The interest coverage ratio, current ratio, and asset-liability ratio belong to solvency. The inventory turnover, current assets turnover, and total assets turnover belong to operating capacity. The revenue growth rate, net profit growth rate, and EPS growth rate belong to growth capacity. The inventory turnover rate belongs to operating capacity. The green emission belongs to Socio-environmental. Therefore, the green credit risk of paper-making enterprises was the result of a combination of two dimensions: credit risk and socio-environmental risk.(5)Based on the Gini index, the interest coverage ratio (X04), current ratio (X02), asset-liability ratio (X01), and green emissions (X19) had been ranked relatively high in terms of their impact on the green credit risk of the paper industry. They can be considered as the main factors affecting the green credit risk of the paper industry.

## 4. Conclusions

This paper extended the application of the random forest model to the field of green credit research. The empirical analysis used 104 samples from 26 paper-making enterprises and established a green credit risk assessment model for the paper industry based on the random forest. The conclusions are as follows: Environment and finance, two main parts of the green credit risk evaluation index system, had a significant impact on assessing the credit risk of the enterprise. On one hand, it was reflected that the green credit risk index system of the paper industry established in this paper had a good reference for green credit risk assessment. Among them, the interest coverage ratio, current ratio, asset-liability ratio, and green emissions had a greater impact on green credit risk in the paper industry. Therefore, commercial banks need to focus on solvency and environmental performance when assessing the green credit risk in the paper industry. On the other hand, the random forest had an excellent performance in green credit risk assessment. The accuracy of sample classification for high-risk enterprises reached 100%. This means that the random forest had an excellent evaluation effect and a high practical value in green credit risk assessment, and can effectively reduce the influence of subjective factors.

The innovation point of this paper mainly involved two aspects. Firstly, the green credit risk index system of the paper industry was constructed with strong industry pertinence, considering both financial and socio-environmental aspects. In particular, the secondary index of green emissions was designed to identify pollution discharge in the paper industry. Secondly, the research method combined the random forest and KMV model. We used the Gini index to continuously select indexes and conducted random forest iterations to establish the model with the best prediction effect. The green credit risk prediction model of the paper industry established in this paper had a high accuracy, which provided some reference for domestic and foreign commercial banks to issue green credit. The development of green credit is an irreversible trend, and thus, we should promote the implementation of green credit to achieve the goal of carbon neutrality as early as possible.

Due to the limited number of Chinese paper enterprises and the polluting emission data of each enterprise not being fully disclosed, the advantages of machine learning with large random forest samples was not the best due to the limitation of sample size. In the future, the model can be adjusted to improve the prediction accuracy by continuously expanding the sample size. Moreover, the green index system in China has not been standardized. If China publishes a complete set of green index systems in the future, the selection of socio-environmental indexes in this paper can be further strengthened.

## Figures and Tables

**Figure 1 ijerph-19-15373-f001:**
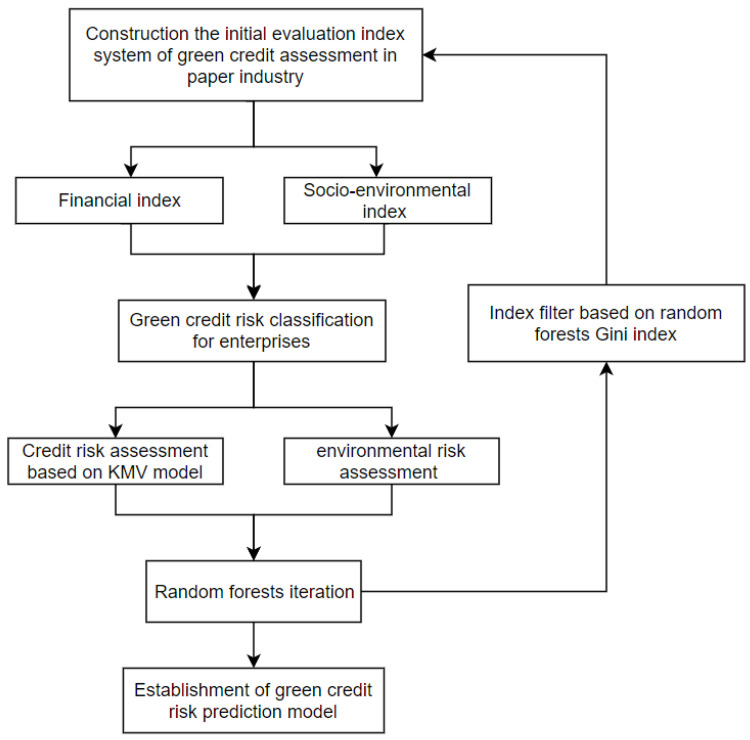
Flow chart of the study.

**Figure 2 ijerph-19-15373-f002:**
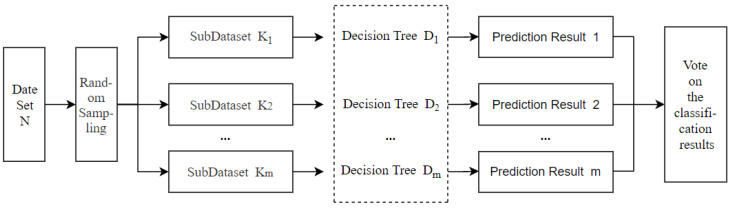
The principle of the random forest algorithm.

**Figure 3 ijerph-19-15373-f003:**
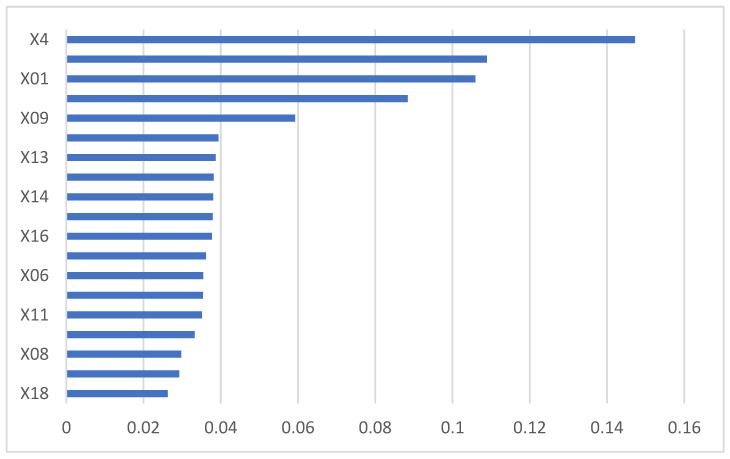
The Gini index of initial indexes.

**Figure 4 ijerph-19-15373-f004:**
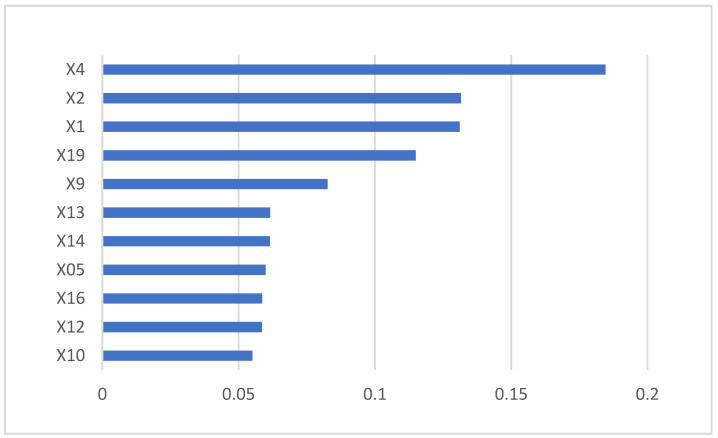
The Gini index of final indexes.

**Table 1 ijerph-19-15373-t001:** Green credit risk evaluation index system.

Perspectives	Primary Index	Secondary Index	Calculation Formula	Number
Financial Index	Solvency	Asset-liability ratio	Total liabilities/total assets	X01
		Current ratio	Current assets/current liabilities	X02
		Quick ratio	Quick assets/current liabilities	X03
		Interest coverage ratio	EBIT/interest expense	X04
	Profitability	ROE	Net profit/net assets	X05
		ROA	Net profit/total assets	X06
		Net asset value per share	Total shareholders’ equity/the number of shares	X07
		Net interest rate in sales	Net profit/main business income	X08
	Operating capacity	Inventory turnover	Operating cost/inventory balance	X09
		Total assets turnover	Operating income/total assets	X10
		Receivable turnover	Operating revenue/accounts receivable balance	X11
		Current assets turnover	Net prime operating revenue/total current assets	X12
	Growth ability	Revenue growth ratio	Revenue growth of the current period/operating income at the start of the period	X13
		Net profit growth ratio	(Net profit of the current period–net profit at the start of the period)/net profit at the start of the period	X14
		Net asset growth ratio	Net profit of the current period/net profit at the start of the period	X15
		EPS growth ratio	Earnings per share of the current period/earnings per share of the prior period	X16
Socio-environmental Index	Social Responsibility	Social responsibility score	Social responsibility score published by Hexun.com (accessed on 26 August 2022)	X17
	Environmental Performance	Green culture	Keyword extraction from annual report or CSR	X18
		Green emissions	The comprehensive score for grading the unit emissions of major pollutants	X19

**Table 2 ijerph-19-15373-t002:** Green emission scores in the paper industry from 2017 to 2021.

Enterprise	2017	2018	2019	2020	2021	Mean	Median	Variance
Chenming Group	0.7805	0.7918	0.8394	0.8397	0.8646	0.8232	0.8313	0.00101
Meili Cloud	0.5124	0.6533	0.8393	0.8120	0.8055	0.7245	0.7650	0.01549
Kan Special Materials	—	0.7990	0.7876	0.8680	0.8915	0.8365	0.8365	0.00196
Jingxing Paper	0.8587	0.8388	0.8118	0.9157	0.9235	0.8697	0.8642	0.00189
Sun Paper	0.8975	0.8778	0.9058	0.9162	0.9276	0.9050	0.9054	0.00029
Hexing Packaging and Printing	—	—	0.9790	0.9703	0.9867	0.9786	0.9788	0.00005
C&S Paper	—	0.8885	0.8952	0.9049	0.9391	0.9069	0.9049	0.00038
Qifeng New Material	—	0.7812	0.7221	0.7000	0.7886	0.7480	0.7480	0.00143
Yuto Packaging Technology	—	—	1.0000	1.0000	0.9999	1.0000	1.0000	0.00000
Yinge Industrial Investment	—	0.8176	0.9495	—	—	0.8836	0.8836	0.00435
Qingshan Paper	—	0.4018	0.3893	0.4041	0.6220	0.4543	0.4041	0.00941
Minfeng Paper	0.5139	0.8564	0.8435	0.7930	0.8272	0.7668	0.8101	0.01644
Huatai Group	0.7594	0.7488	0.8232	0.8534	0.8430	0.8055	0.8144	0.00187
Hengfeng Paper	0.9057	0.8874	0.8771	0.9091	0.9147	0.8988	0.9023	0.00020
Guanhao High-Tech	0.9372	0.9711	0.9521	0.9689	0.9198	0.9498	0.9510	0.00038
Shanying Intl.	0.8089	0.9292	0.8895	0.8924	0.9146	0.8869	0.8910	0.00174
Yibin Paper	0.3984	0.8296	0.8331	0.6927	0.7083	0.6924	0.7005	0.02506
Yueyang Forest & Paper	0.3717	0.6893	0.7205	0.7644	0.8596	0.6811	0.7049	0.02722
Bohui Paper	0.9200	0.8567	0.7701	0.7476	0.8517	0.8292	0.8405	0.00393
Rong Sheng	—	0.7872	0.8599	0.8495	0.8633	0.8400	0.8495	0.00095
Jinghua Laser	—	—	0.8141	0.8110	0.8291	0.8181	0.8161	0.00006
Xianhe Corp.	—	0.8945	0.9019	0.8769	0.8941	0.8919	0.8941	0.00008
Songyang Recycle Resources	—	—	0.8303	0.8653	0.4946	0.7301	0.7802	0.02792
Wuzhou Special Paper Group	—	—	—	0.7506	0.7126	0.7316	0.7316	0.00036
Huawang	—	—	—	0.8366	0.8926	0.8646	0.8646	0.00078
Forest Packing	—	—	—	0.8723	0.8558	0.8641	0.8641	0.00007

— indicates the enterprise is not listed or the data on pollutant emissions are missing.

**Table 3 ijerph-19-15373-t003:** Default distances of listed enterprises in the paper industry from 2017 to 2021.

Enterprise	2017	2018	2019	2020	2021	Mean	Median	Variance
Chenming Group	−1.7002	−4.0146	−7.9213	−4.3010	−2.0063	−3.9887	−4.0017	4.9461
Meili Cloud	2.2244	1.4740	1.3757	1.8407	1.8038	1.7437	1.7738	0.0905
Kan Special Materials	—	1.3629	1.4217	1.8999	2.2633	1.7370	1.7370	0.1357
Jingxing Paper	2.7420	1.3958	2.6275	1.6898	2.0180	2.0940	2.0560	0.2722
Sun Paper	1.8706	0.4159	1.4312	1.5428	1.0923	1.2706	1.3509	0.2444
Hexing Packaging and Printing	—	—	1.5237	1.0230	0.3177	0.9548	0.9889	0.2447
C&S Paper	—	1.2970	2.5877	2.2795	2.0742	2.0596	2.0742	0.2273
Qifeng New Material	—	3.2025	2.6165	1.8979	1.4197	2.2841	2.2841	0.4626
Yuto Packaging Technology	—	—	0.6495	1.4944	1.9341	1.3593	1.4269	0.2842
Yinge Industrial Investment	—	2.1423	0.9667	—	—	1.5545	1.5545	0.3455
Qingshan Paper	—	0.8499	2.2591	2.3239	1.9763	1.8273	1.9763	0.3520
Minfeng Paper	2.4164	1.3532	1.2522	1.7351	1.6347	1.6819	1.6583	0.1674
Huatai Group	1.2538	0.4599	0.8748	0.7758	0.7100	0.8149	0.7954	0.0670
Hengfeng Paper	3.2235	2.5738	1.7861	2.6007	2.4578	2.5284	2.5511	0.2093
Guanhao High-Tech	2.8822	1.7373	2.4830	2.3974	1.6237	2.2247	2.3111	0.2255
Shanying Intl.	1.4617	−0.0638	−0.1390	−1.1905	−1.3718	−0.2445	−0.1918	1.0239
Yibin Paper	0.0708	−1.1236	0.0708	0.1500	−0.3031	−0.1632	−0.0462	0.2258
Yueyang Forest & Paper	0.4915	−0.4587	−0.2268	0.7914	1.0819	1.6793	0.6415	0.3472
Bohui Paper	0.6240	−3.5682	−1.1080	0.9563	1.7231	−0.2746	0.1747	3.5712
Rong Sheng	—	1.3070	1.8010	1.9239	2.4844	1.8791	1.8791	0.1754
Jinghua Laser	—	—	1.8686	1.8206	2.6086	2.0993	1.9840	0.1301
Xianhe Corp.	—	−1.4424	−4.1023	1.0849	2.0169	−0.6107	−0.6107	5.6656
Songyang Recycle Resources	—	—	2.0513	1.6367	1.8536	1.8472	1.8504	0.0287
Wuzhou Special Paper Group	—	—	—	−0.8119	0.2918	−0.2600	−0.2600	0.3045
Huawang	—	—	—	1.4342	1.0295	1.2318	1.2318	0.0409
Forest Packing	—	—	—	5.1820	2.4876	3.8348	3.8348	1.8149

— indicates the enterprise is not listed or the data on pollutant emissions are missing.

**Table 4 ijerph-19-15373-t004:** Initial parameters with the minimum misclassification rate.

Predicted Value/True Value	(X = 19 m_features = 0.4)		Misclassification Rate
	Risk-Free	Risky	
Risk-free	14	3	12.50%
Risky	1	14	

**Table 5 ijerph-19-15373-t005:** Final parameters with the minimum misclassification rate.

Predicted Value/True Value	(X = 11 m_features = 0.2)		Misclassification Rate
	Risk-Free	Risky	
Risk-free	21	0	6.25%
Risky	2	9	

**Table 6 ijerph-19-15373-t006:** Main influencing indicators of green credit risk under different parameters.

Index	Order of Importance	
	i = 1	i = 0.5
Interest coverage ratio	1	1
Current ratio	2	3
Asset-liability ratio	3	4
Green emission	4	2
Inventory turnover	5	5
Current assets turnover	6	7
Revenue growth ratio	7	12
Quick ratio	8	5
Net profit growth ratio	9	11
Total assets turnover	10	10
EPS growth ratio	11	7

## Data Availability

The data analyzed in this study are subject to the following licenses/restrictions: Belong to Study Group Requirements. Requests to access these datasets should be directed to Y.C., sanchen007@njfu.edu.cn.

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
