# Peer review of "Assessing and Predicting Green Credit Risk in the Paper Industry"

_ijerph, 2022, doi:10.3390/ijerph192215373_

Round 1

Reviewer 1 Report

- In the first paragraph of the Introduction, please clarify what "Green Trade Barriers" are.

- Your paper would benefit from indicating the findings in the literature about the value relevance of corporate environmental performance. For example, Arslan-Ayaydin and Thewissen (2015) shows that reducing environmental concerns pays off by improved corporate profitability. On the other hand, Iwata and Okada (2011) show different effects of each environmental performance on financial performance.

Arslan-Ayaydin, Ö., & Thewissen, J. (2015). The impact of environmental strengths and concerns on the accounting performance of firms in the energy sector. In Energy Technology and Valuation Issues (pp. 83-107). Springer, Cham.  

Iwata, H., & Okada, K. (2011). How does environmental performance affect financial performance? Evidence from Japanese manufacturing firms. Ecological Economics, 70(9), 1691-1700.

- Please briefly explain the Hexun.com

- Please justify why you measure "Net interest rate in sale" as "Net Profit/ Operating Income".

- Please provide a clear explanation of the KMV model.

- What is the total number of A-share listed paper-making enterprises?

- Rather than indicating the individual names of each company on Tables 2 and Table 3, please indicate the mean and median values of the all sample between 2017 and 2021.

- Please remove Section 3.2.3,

- Please edit your text. For example, the following first sentence of the conclusion is very poorly written;

"This paper extends the application of the random forest classification algorithm to the field of green credit research, establishes a green credit risk assessment and prediction model for the paper industry based on the random forest, which collects 104 sample data 438 from 26 paper industries for empirical analysis, the conclusion is as follows."

Author Response

Response to Reviewer 1 Comments

Response to Reviewer 1 Comments for

International Journal of Environmental Research and Public Health - 1963457

“Assessing and Predicting the Green Credit Risk in the Paper Industry”

Point 1: In the first paragraph of the Introduction, please clarify what "Green Trade Barriers" are.

Response 1: Thank you for the suggestions, which are very helpful for improving the quality of our manuscript. Green trade barrier refers to a series of import restriction measures formulated by the importing country to protect natural resources, ecological environment and human health in the international trade activities.  It mainly includes international and regional environmental protection conventions, national environmental protection regulations and standards, ISO14000 environmental management system, and production and processing method. If the technical of green papermaking is relatively low, it is easy to suffer from green trade barriers, which is not conducive to its future development. We have added the explanation of "Green Trade Barriers" in the first paragraph of the Introduction. Please see Line 34-36 for details.

Point 2: Your paper would benefit from indicating the findings in the literature about the value relevance of corporate environmental performance. For example, Arslan-Ayaydin and Thewissen (2015) shows that reducing environmental concerns pays off by improved corporate profitability. On the other hand, Iwata and Okada (2011) show different effects of each environmental performance on financial performance.

Response 2: Thank you for the suggestions, which are very helpful for improving the quality of our manuscript. The first literature indicated that profitability is linked to environmental concerns, thus it was added in the first paragraph of Section 2.1. Please see Line 146-148 for details. The second literature use the major pollution emissions of the industry as a proxy for environmental performance, thus it was added in the fourth paragraph of Section 2.1. Please see Line 189-190 for details.

Point 3: Please briefly explain the Hexun.com.

Response 3: Thank you for the comments. Hexun Social Responsibility Score is the score that Hexun systematically evaluates the social responsibility of listed enterprises through the establishment of relevant index system and the score will be published on Hexun.com. We have added a brief explanation of Hexun.com in the third paragraph of Section 2.1. Please see Line 171-174 for details.

Point 4: Please justify why you measure "Net interest rate in sale" as "Net Profit/ Operating Income".

Response 4: Thank you for the suggestions, which are very helpful for improving the quality of our manuscript. Operating income should be changed to Main business income, that is, Net interest rate in sale = Net profit / Main business income. It was an error in the translation, which has been corrected in the Table 1. In addition, we also conducted a new round of inspection to make sure the formula is correct. 

Point 5: Please provide a clear explanation of the KMV model.

Response 5: Thank you for the comments. We have revised the Introduction and the Discussion of Results (Section 3.4). Firstly, adding a brief introduction to the KMV model and providing more literatures on KMV model for corporate credit risk and government debt risk assessment. Please see Line 60-73 for details. Secondly, adding results discussion related to the KMV model in the Section 3.4. Discussion of Results (1) summarizes the credit risk situation of paper industry. Please see Line 447-451 for details.

Point 6: What is the total number of A-share listed paper-making enterprises?

Response 6: Thank you for the comments, which are very helpful for improving the quality of our manuscript. The total number of A-share listed paper-making enterprises in China is 40. There is no mandatory requirement to publish enterprise pollution emission data before 2021, so the data publication is not comprehensive. This paper screened out the paper-making enterprises which have serious missing of environmental information disclosure, and the remaining is 26. We have added an explanation in the Section 2.4. Please see Line 306-310 for details.

Point 7: Rather than indicating the individual names of each company on Tables 2 and Table 3, please indicate the mean and median values of the all sample between 2017 and 2021.

Response 7: Thank you for the comments, which are very helpful for improving the quality of our manuscript. We have added a column of mean and median values on Tables 2 and Table 3. After careful consideration, we decided to keep the individual names of each company for the following analysis.

Point 8:  Please remove Section 3.2.3

Response 8: Thank you for the suggestions, which are very helpful for improving the quality of our manuscript. The table 4 in the section 3.2.3 gives the results of enterprise green credit risk classification. After careful consideration, we decided to delete this section from the manuscript by the following reasons. Firstly, the classification basis has been specified above, and deleting this section will not affect my research framework. Secondly, the length of the section 3.2.3 is too short and the section 3.2.3 has no complex calculation.

Point 9:  Please edit your text. For example, the following first sentence of the conclusion is very poorly written; "This paper extends the application of the random forest classification algorithm to the field of green credit research, establishes a green credit risk assessment and prediction model for the paper industry based on the random forest, which collects 104 sample data 438 from 26 paper industries for empirical analysis, the conclusion is as follows."

Response 9: Thank you for the comments. We have made proof read and manuscript again and to correct many grammar mistakes and syntax mistakes. In addition, we also conducted a new round of polishing on manuscript to increase the readability of our paper. We will continue to improve our works in terms of both English language usage and quality of contents. For example, the first paragraph of the conclusion has been proved. Please see Line 480-485 for details.

Reviewer 2 Report

Only 1 of your 3 innovative indexes are listed in the final model. You are not discussing this strange result that could make the full paper and the research questions not relevant.

The table 4 seems very strange because for some years you have more than 50% of companies classified as riskier and exposed to environmental penalty... normally in standard risk models the probability of an event related to default has a probability of 5% or lower. Please explain why you sample is suffering of such type of problems.

Literature on the topic of credit risk modelling and KMV is not analysed in detail and only few references are provided

The paper has not robustness test and results cannot be generalized

Author Response

Response to Reviewer 2 Comments

Response to Reviewer 2 Comments for

International Journal of Environmental Research and Public Health - 1963457

“Assessing and Predicting the Green Credit Risk in the Paper Industry”

Point 1: Only 1 of your 3 innovative indexes are listed in the final model. You are not discussing this strange result that could make the full paper and the research questions not relevant.

Response 1: Thank you for the comments, which are very helpful for improving the quality of our manuscript. The three innovative indexes are set from the perspective of the degree of corporate social responsibility (Social Responsibility Score), environmental awareness (Green Culture) and actual environmental protection actions (Green Emission).  The redundancy of indicators will reduce the accuracy and operation speed of the model. Thus, this paper uses the random forest iteration method to find the index system with the highest prediction accuracy. After random forest iteration, Green Emission is left only, which means that enterprises' actual environmental protection actions play a more important role in the classification of green credit risk. We have added the explanation in the Section 3.3.1. Please see Line 421-426 for details.

Point 2: The table 4 seems very strange because for some years you have more than 50% of companies classified as riskier and exposed to environmental penalty... normally in standard risk models the probability of an event related to default has a probability of 5% or lower. Please explain why you sample is suffering of such type of problems.

Response 2: Thank you for the comments, which are very helpful for improving the quality of our manuscript. The reasons for the high default ratio of green credit in the sample of this paper as follows. 

  • The table 4 shows the classification results of green credit risk, which not only considers credit risk but also environmental punishment. The classification requirements for low risk of green credit in this paper are that there is neither credit risk nor environmental punishment, which will be greater than simply considering the probability of default of enterprise credit risk. Therefore, there are more high-risk enterprises.
  • The paper industry has become a key credit restricted industry of China's commercial banks because of the overcapacity. While facing the dual pressures of economic downturn and structural adjustment, the business environment is becoming more difficult, that is the reason why the credit risk of the industry is rising and the default probability of paper enterprises is higher.

     The table 4 in the section 3.2.3 gives the results of enterprise green credit risk classification. After careful consideration, we decided to delete this section from the manuscript by the following reasons. Firstly, the classification basis has been specified above, and deleting this section will not affect my research framework. Secondly, the length of the section 3.2.3 is too short and the section 3.2.3 has no complex calculation.

Point 3: Literature on the topic of credit risk modelling and KMV is not analysed in detail and only few references are provided.

Response 3: Thank you for the comments. We have revised the Introduction. Firstly, adding a brief introduction to the KMV model. Please see Line 60-62 for details. Secondly, adding the literatures about corporate credit risk and government debt risk assessment based on KMV model. Please see Line 62-73 for details. Thirdly, Adding the literatures on different methods to assessing credit risk in the Introduction. Please see Line 76-83 for details.

Point 4: The paper has not robustness test and results cannot be generalized.

Response 4: Thank you for the comments, which are very helpful for improving the quality of our manuscript. A new section (3.3.2 Robustness Test) has been added. Based on the actual of green credit risk exposure in the paper industry, this paper assumes that the DD is less than 1 or that enterprises subject to environmental penalties have potential green credit risks. In order to further test the robustness of the model, this assumption is now adjusted, that is, the DD is less than 0.5 or the enterprises subject to environmental penalties have potential credit risks. The results show that when the misclassification rate of the random forest drops to a low level, the final indexes of the model include interest coverage ratio, current ratio, asset-liability ratio, green emission, inventory turnover, current assets turnover, revenue growth ratio, quick ratio, net profit growth ratio, total assets turnover, EPS growth ratio and other indicators, but the order of importance is slightly adjusted. It can be concluded that the model has good robustness. Please see Line 429-445 for details.

Round 2

Reviewer 2 Report

the paper was revised accordingly with the review report